# Epipharyngeal Abrasive Therapy (EAT) Reduces the mRNA Expression of Major Proinflammatory Cytokine IL-6 in Chronic Epipharyngitis

**DOI:** 10.3390/ijms23169205

**Published:** 2022-08-16

**Authors:** Kensuke Nishi, Shohei Yoshimoto, Soichiro Nishi, Tatsuro Nishi, Ryushiro Nishi, Takayuki Tanaka, Toshiyuki Tsunoda, Kazuaki Imai, Hiroaki Tanaka, Osamu Hotta, Ayaki Tanaka, Kenji Hiromatsu, Senji Shirasawa, Takashi Nakagawa, Takafumi Yamano

**Affiliations:** 1Section of Otolaryngology, Department of Medicine, Fukuoka Dental College, Fukuoka 814-0193, Japan; 2Nishi Otolaryngology Clinic, Fukuoka 814-0031, Japan; 3Department of Otolaryngology, Faculty of Medicine, Fukuoka University, Fukuoka 814-0180, Japan; 4Section of Pathology, Department of Morphological Biology, Division of Biomedical Sciences, Fukuoka Dental College, Fukuoka 814-0193, Japan; 5Oral Medicine Research Center, Fukuoka Dental College, Fukuoka 814-0193, Japan; 6Department of Cell Biology, Faculty of Medicine, Fukuoka University, Fukuoka 814-0180, Japan; 7Mirai Clinic, Fukuoka 812-0013, Japan; 8Tanaka Hiroaki Clinic, Fukuoka 814-0142, Japan; 9Division of Internal Medicine, Hotta Osamu Clinic (HOC), Sendai 984-0013, Japan; 10Tanaka ENT Clinic, Osaka 553-0006, Japan; 11Department of Microbiology and Immunology, Faculty of Medicine, Fukuoka University, Fukuoka 814-0180, Japan; 12Department of Otorhinolaryngology, Graduate School of Medical Sciences, Kyushu University, Fukuoka 812-8582, Japan

**Keywords:** epipharyngeal abrasive therapy (EAT), interleukin 6 (IL-6), IgA nephropathy, myalgic encephalomyelitis/chronic fatigue syndrome (ME/CFS), long COVID, tumor necrosis factor alpha (TNFα)

## Abstract

The epipharynx, located behind the nasal cavity, is responsible for upper respiratory tract immunity; however, it is also the site of frequent acute and chronic inflammation. Previous reports have suggested that chronic epipharyngitis is involved not only in local symptoms such as cough and postnasal drip, but also in systemic inflammatory diseases such as IgA nephropathy and myalgic encephalomyelitis/chronic fatigue syndrome (ME/CFS) and Long COVID. Epipharyngeal Abrasive Therapy (EAT), which is an effective treatment for chronic epipharyngitis in Japan, is reported to be effective for these intractable diseases. The sedation of chronic epipharyngitis by EAT induces suppression of the inflammatory cytokines and improves systemic symptoms, which is considered to be one of the mechanisms, but there is no report that has proved this hypothesis. The purpose of this study was to clarify the anti-inflammatory effect of EAT histologically. The study subjects were 8 patients who were not treated with EAT and 11 patients who were treated with EAT for chronic epipharyngitis for 1 month or more. For immunohistochemical assessment, the expression pattern of IL-6 mRNA, which plays a central role in the human cytokine network, was analyzed using in situ hybridization. The expression of IL-6 in the EAT-treated group was significantly lower than those in the EAT nontreated group (*p* = 0.0015). In addition, EAT suppressed the expression of tumor necrosis factor alpha (TNFα), a crucial proinflammatory cytokine. As a result, continuous EAT suppressed submucosal cell aggregation and reduced inflammatory cytokines. Thus, EAT may contribute to the improvement of systemic inflammatory diseases through the suppression of IL-6 expression.

## 1. Introduction

The epipharynx, which has well-developed submucosal lymphoid tissue, is responsible for host defense mucosal immunity, but it is also susceptible to local inflammation due to upper respiratory tract infection (URTI), air pollution, and pollen [1,2]. The ciliated structure of the epipharyngeal mucosa highly expresses various virus entry factors, suggesting the importance of the epipharynx in URTI [3,4]. Epipharyngitis is involved not only in otolaryngological disorders, but also in the development of systemic diseases such as IgA nephropathy, palmoplantar pustulosis, rheumatic arthritis, psoriasis, myalgic encephalomyelitis/chronic fatigue syndrome (ME/CFS), and Long COVID [1,5,6,7,8,9]. We have previously shown endoscopically that patients with Long COVID have residual epipharyngeal inflammation [5]. Epipharyngeal abrasive therapy (EAT), which is a treatment for chronic epipharyngitis in Japan, has been reported to be effective in improving these pathological conditions [5,10,11]. It has been pointed out that the anti-inflammatory effect of ZnCl_2_ applied to cotton swabs used in EAT with a sufficient abrasive procedure inhibit the epipharyngeal-activated lymphocytes [1,2], but the underlying mechanism has not been elucidated. The inflammatory cytokine IL-6 is highly expressed in various chronic inflammatory diseases and is a therapeutic target in inflammatory diseases [12,13]. In fact, the high expression level of IL-6 in serum or plasma has been reported to be involved in the development of IgA nephropathy, ME/CFS, and Long COVID [14,15,16,17,18]. The purpose of this study was to clarify the mechanism of effectiveness for systemic diseases of EAT by analyzing the expression pattern of IL-6 in the epipharyngeal mucosa of patients with chronic epipharyngitis before and after EAT. We revealed that EAT reduces the expression of IL-6 mRNA by preventing the collection of inflammatory cells in the epipharynx. Our results may provide evidence that EAT is effective in a variety of disorders caused by chronic epipharyngitis.

## 2. Results

### 2.1. Epipharyngeal Abrasive Therapy (EAT) Improved Chronic Epipharyngitis Endoscopically

To confirm the effectiveness of EAT for chronic epipharyngitis, we analyzed the endoscopic changes in a patient before and after EAT (Figure 1). Severe mucosal swelling, cobblestone-like granular changes, and submucosal bleeding confirmed before EAT are each characteristic mucosal findings of chronic epipharyngitis. Continuous EAT reduced mucosal swelling, and the blood vessels under the epipharyngeal mucosa visualized in green could be confirmed in narrow band imaging (NBI) mode. EAT induced a temporary whitening phenomenon, which is considered to indicate reduced inflammation.

### 2.2. B Cells Mainly Secrete Interleukin 6 (IL-6) in the Epipharynx

To investigate the secretion cells of IL-6 in the epipharynx, double staining was performed with IL-6 mRNA and each cell marker (Figure 2). Both B cells and T cells were present in the lymphoid tissue of the epipharynx of patients with chronic nasopharyngitis, and macrophages were also present mainly under the mucosa. IL-6 was mainly co-localized with B cells and was less expressed on T cells. No IL-6 expression was observed in macrophages and vascular endothelial cells. These results showed that B cells mainly secrete IL-6 in the lymphoid tissue of the epipharynx.

### 2.3. Epipharyngeal Abrasive Therapy (EAT) Downregulates the Expression of Interleukin 6 (IL-6)

To clarify histologically whether EAT suppresses inflammation in correlation with macroscopic endoscopic findings, we investigated the changes in the expression of the inflammatory cytokine IL-6 mRNA before and after EAT. IL-6 was highly expressed in the submucosal region, where many lymphocytes were gathered in the EAT nontreated group, whereas the expression of IL-6 was reduced with a decrease in lymphocyte aggregation in the EAT-treated group (Figure 3a). Figure 3b and Table 1 show the presence or absence of EAT and IL-6 expression in patients with chronic epipharyngitis. As a result of the chi-square test, a significant difference in IL-6 expression was obtained between the epipharynx without and with EAT (*p* = 0.0015). These results indicated that EAT improved chronic epipharyngitis histologically.

### 2.4. Epipharyngeal Abrasive Therapy (EAT) Downregulates the Expression of Tumor Necrosis Factor Alpha (TNFα)

To clarify whether EAT affects the expression of typical inflammatory cytokines other than IL-6, we investigated the changes in the expression of TNFα mRNA before and after EAT. TNFα was highly expressed at the submucosal region in the EAT nontreated group, whereas the expression of TNFα was reduced in the EAT-treated group (Figure 4). This result is similar to the decrease in the IL-6 expression caused by EAT.

## 3. Discussion

The epipharynx is a physiologically immunoactive site, which serves as a source of antigen-specific IgA-producing B cells responsible for protective mucosal immunity [19,20], but hyperinflammation of the epipharynx is associated with not only local symptoms, but also the development of intractable systemic diseases such as IgA nephropathy (IgAN), myalgic encephalomyelitis/chronic fatigue syndrome (ME/CFS), and Long COVID [5,6,7,21]. The excessive secretion of inflammatory cytokines associated with epipharyngitis is considered to be one of the causes, and Epipharyngeal Abrasive Therapy (EAT), which is a treatment method for chronic epipharyngitis in Japan, effectively improves these pathologies [5,10,22]. EAT reduces inflammation by scratching the entire epipharyngeal mucosa with a cotton swab soaked in zinc chloride, which has an anti-inflammatory effect (Figure 1) [1]. It has been reported that EAT improves not only local pharyngeal symptoms, but also various systemic symptoms by suppressing inflammation of the epipharynx [6]. This effectiveness of EAT is also supported by the effectiveness of the excision of the palatine tonsils, which have epipharyngeal-like lymphatic tissue, to improve autoimmune-related symptoms [19,23]. Although inflammation sedation by EAT is inferred from macroscopic findings using an endoscope [5,24], the histological changes in local immunity of the epipharynx due to EAT were unknown.

This study focused on IL-6, which plays a central role in the human cytokine network, and histologically investigated the effect of EAT on the IL-6 expression in the epipharynx. IL-6 is a multifaceted cytokine with various biological activities in immunomodulation, which is produced in infectious or injured lesions and delivered systemically via the bloodstream to drive the host defense system [25]. On the other hand, excessive IL-6 secretion induces chronic inflammatory diseases, autoimmune diseases, cancer, and cytokine storms [26,27]. Therefore, treatment strategies focused on the IL-6 pathway have been researched in various areas such as cancer, inflammation, and autoimmune diseases; in fact, anti-IL6 therapy is available for Castleman disease and rheumatoid arthritis. [13,28].

In this study, we attempted to identify IL-6-secreting cells in the epipharynx using cytokine RNA in situ hybridization with a high degree of specificity [29]. We found that B cells present in the submucosal lymphoid tissue mainly express IL-6 mRNA in the epipharyngeal mucosa of patients with chronic epipharyngitis (Figure 2). IL6 secretion from B cells correlates with the severity of humoral autoimmune disease [30], and IL6 is involved in the pathogenesis of IgAN by promoting IgA production [17,31]. In addition, sustained IL-6 activity also affects the development of ME/CFS and neuropsychiatric symptoms of Long COVID [14,18,32]. It is thought that one of the causes is that IL-6, which is an inflammatory cytokine, crosses the blood–brain barrier and affects the energy metabolism of neurons [18,33]. Interestingly, we revealed that continuous EAT suppresses submucosal cell aggregation and reduces the mRNA expression of inflammatory cytokine IL-6 (Figure 3). Since mRNA expression of IL-6 in lymphoid tissues correlates with local protein expression [34,35], the reduced mRNA expression associated with EAT is assumed to suppress IL-6 secretion in the epipharynx. This study is the first report to histologically clarify the anti-inflammatory effect of EAT. Regarding the mechanism of decreased cell aggregation, we previously reported that it was due to fibrotic stroma formation of submucosal tissue by EAT. Since scratching with a cotton swab soaked in zinc chloride is important for the symptom improvement effect of EAT, it is speculated that not only the protein denaturing effect of zinc chloride, but also the mechanical irritation caused by scratching is involved in this histological change.

In this study, we clarified the suppression of IL-6 expression locally in the epipharynx using EAT, and the results indicate that EAT may be effective against various chronic inflammatory diseases involving IL6. IL6 is also involved in the pathophysiology of allergic diseases, and high levels of IL-6 in the blood promote the release of allergic mediators and exacerbate allergic rhinitis [36,37]. In addition, it has been suggested that the allergic rhinitis symptoms are alleviated by inhibiting the higher IL-6 expression level [38], which is a result supporting the pharyngeal allergic inhibitory effect of EAT [2]. Similar to IL-6, we showed that EAT downregulates the mRNA expression of tumor necrosis factor alpha (TNFα), one of the most crucial proinflammatory cytokines, along with IL-6 (Figure 4). TNFα is produced mainly by activated macrophages, T cells and B cells, and induces the secretion of various cytokines and chemokines [39,40]. TNFα is involved in a variety of biological activities for cell survival, differentiation, and proliferation, but chronic inflammation associated with excessive production of TNFα may lead to the development of autoimmune diseases such as rheumatoid arthritis, psoriasis, noninfectious uveitis, and inflammatory bowel disease [41]. Thus, TNFα has been studied as a therapeutic target for intractable diseases, and several TNF-α Inhibitors have been approved as therapeutic agents for autoimmune diseases [42,43,44,45]. These TNFα-related diseases have been reported to be associated with chronic epipharyngitis, and EAT effectively improves symptoms in some cases [1,6], suggesting that the suppression of TNFα expression in the epipharynx associated with EAT may contribute to the improvement of systemic inflammatory diseases, along with the suppression of IL-6 expression.

We have not confirmed any systemic change including blood cytokine levels via local inflammation improvement by EAT in this research, and further studies are required. Nevertheless, the chronic inflammation of epipharynx, which is responsible for upper respiratory tract immunity along with palatal tonsils, may induce systemic disease through a mechanism similar to tonsil-induced autoimmune/inflammatory syndrome (TIAS) [23,46]. We histologically substantiated the hypothesis that EAT subdues local epipharyngeal inflammation. As a result, we speculate that the reduction in IL-6, an important mediator of the inflammatory pathway, is one of the mechanisms by which EAT improves systemic disease.

## 4. Materials and Methods

### 4.1. Patients and Tissue Samples

Patient information was obtained with permission from the Ethics Committee of Fukuoka Dental College (ID: 552). Samples were obtained by endoscopic epipharyngeal biopsy between July 2021 and August 2021. The study subjects were 8 patients who had not been treated with EAT and 11 patients who had been treated with EAT for chronic epipharyngitis for 1 month or more, as described previously [3]. All participants provided informed written consent. The research was conducted in accordance with the Declaration of Helsinki and Title 45, US Code of Federal Regulations, Part 46, Protection of Human Subjects, effective as of 13 December 2001.

### 4.2. Epipharyngeal Abrasive Therapy (EAT)

EAT is a treatment method for chronic epipharyngitis that has been performed using cotton swabs soaked in ZnCl_2_ in Japan since the 1960s. EAT was performed in the Outpatient Department one to three times a week by a Board-certified otorhinolaryngologist, as described previously (Figure 5) [3].

### 4.3. Antibody

Mouse antibodies to CD20 (clone: L26), CD3 (clone: F7.2.38), and CD68 (clone: PG-M1) were purchased from DAKO-Agilent Technologies Co., Ltd. (Santa Clara, CA, USA). Mouse antibody to CD34 (clone: NU-4A1) was purchased from Nichirei Biosciences (Tokyo, Japan). The secondary horseradish peroxidase (HRP)-conjugated polymer anti-rabbit antibody was purchased from DAKO-Agilent Technologies Co., Ltd. (Santa Clara, CA, USA).

### 4.4. RNA in Situ Hybridization (ISH) and Immunohistochemical (IHC) Staining

Neutral buffered formalin (10%)-fixed and paraffin-embedded tissue blocks were cut into 4 μm-thick sections for ISH and IHC staining. To detect IL-6 and TNF-α mRNA on the tissue, an RNA scope (in situ hybridization system, Advanced Cell Diagnostics, Hayward, CA, USA; IL-6: No. 310371, TNF-α: No. 310421) was used following the manufacturer’s guidelines. The positivity of signals in the subepithelial region was determined by pathologists at high magnification. Briefly, we randomly chose three areas at the subepithelial region, determined based on at least 300 cells. In the dual staining of ISH and IHC, the sections after ISH were treated with 5% bovine serum albumin/Tris-buffered saline to block any nonspecific binding of primary antibodies. Subsequently, each section was incubated with the primary antibody against CD20, CD3, CD68, or CD34 at 4 °C overnight. These sections were then incubated with HRP-conjugated polymer anti-rabbit or anti-mouse antibody. The peroxidase activity was visualized using 0.1% 3,3′-diaminobenzidine and 0.01% hydrogen peroxide in Tris-buffered saline. All images were captured using a microscope (AXIO Vert.A1, Carl Zeiss Inc., Carl Zeiss, Oberkochen, Germany).

### 4.5. Statistical Analysis

Histopathological evaluation results were statistically analyzed using the chi-square test with Graph-Pad Prism (version 9.2.0 for Windows; GraphPad Software, San Diego, CA, USA).

## 5. Conclusions

EAT reduces the mRNA expression of IL-6, a key cytokine in chronic inflammation, in the epipharynx, suggesting that EAT may contribute to the improvement of systemic inflammatory diseases involving IL-6.

## Figures and Tables

**Figure 1 ijms-23-09205-f001:**
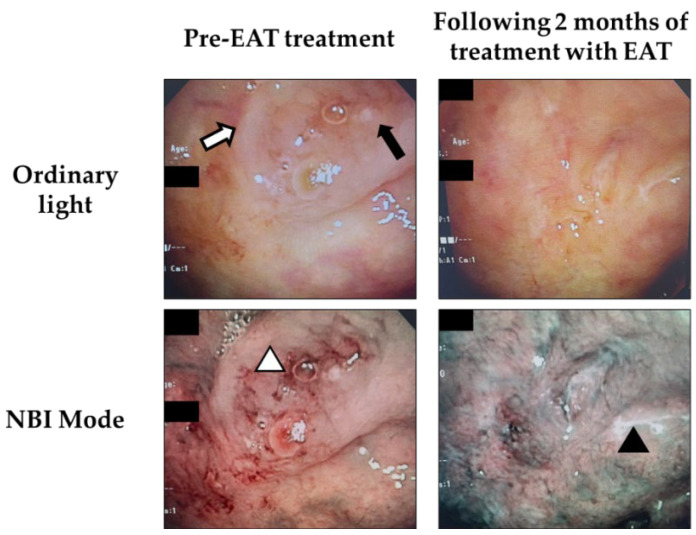
Transnasal endoscopic photographs of the epipharynx under ordinary light and Narrow Band Imaging (NBI) mode in a patient with chronic epipharyngitis. (**Left**) panel shows the epipharynx pre-EAT treatment. (**Right**) panel shows the epipharynx following 2 months of treatment with EAT. The white arrow indicates severe mucosal swelling. The black arrow indicates cobblestone-like granular changes. The white arrowhead indicates submucosal bleeding. The black arrowhead indicates the temporary whitening phenomenon.

**Figure 2 ijms-23-09205-f002:**
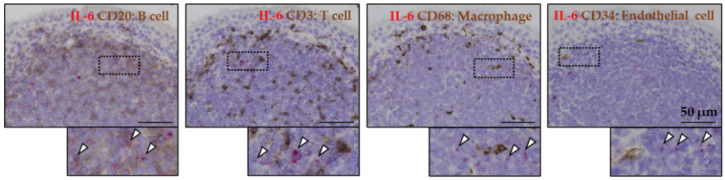
The distribution of Interleukin 6 (IL-6) mRNA (red dots; arrow heads), B cells (CD20+), T cells (CD3+), macrophages (CD68+), and vascular endothelial cells (CD34+) in the epipharynx of a patient with chronic epipharyngitis. Inserts in each image are magnified.

**Figure 3 ijms-23-09205-f003:**
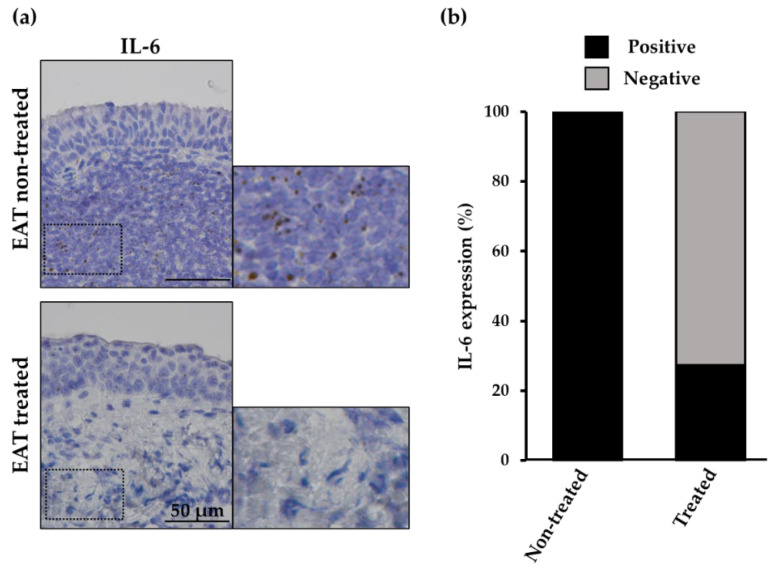
The mRNA expression patterns of Interleukin 6 (IL-6) in patient tissue samples without and with epipharyngeal abrasive therapy (EAT). (**a**) The representative pattern of IL-6 expression (brown dots) in the epipharynx of the EAT nontreated sample and the EAT-treated sample. Inserts in each image are magnified. (**b**) The details of IL-6 expression at the submucosal region of the EAT nontreated group (*n* = 8) and EAT-treated group (*n* = 11).

**Figure 4 ijms-23-09205-f004:**
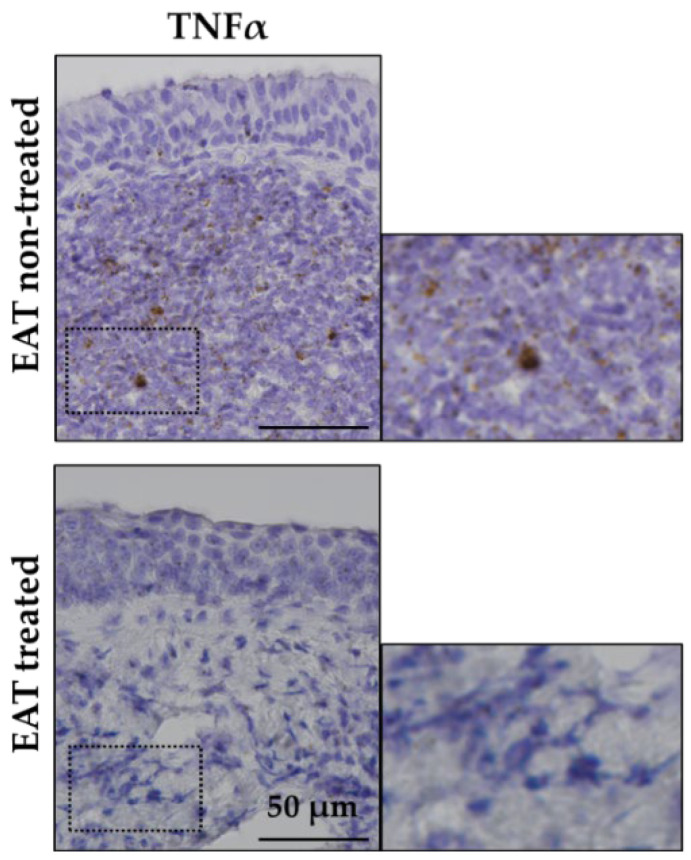
The mRNA expression patterns of tumor necrosis factor alpha (TNFα) in patient tissue samples without and with epipharyngeal abrasive therapy (EAT). TNFα (brown dots) in the epipharynx of the EAT nontreated sample (*n* = 1) and the EAT-treated sample (*n* = 1). Inserts in each image are magnified.

**Figure 5 ijms-23-09205-f005:**
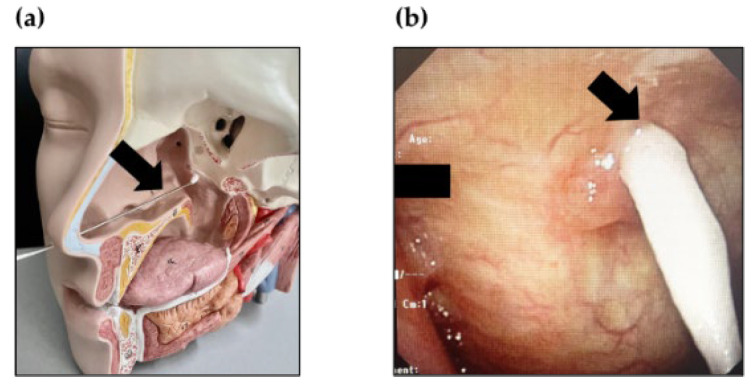
Epipharyngeal Abrasive Therapy (EAT). (**a**) The transnasal EAT technique using a 3D model (Nihon 3B Scientific Inc., Niigata, Japan). (**b**) The epipharynx during endoscopic EAT (E-EAT). The black arrow indicates a cotton swab.

**Table 1 ijms-23-09205-t001:** Chi-square test for the expression of Interleukin 6 (IL-6) before and after EAT.

	IL-6 Positive	IL-6 Negative	Chi-Square Value
EAT nontreated	8	0	0.0015
EAT-treated	3	8	

## Data Availability

Not applicable.

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
