# Peer review of "Epipharyngeal Abrasive Therapy (EAT) Reduces the mRNA Expression of Major Proinflammatory Cytokine IL-6 in Chronic Epipharyngitis"

_ijms, 2022, doi:10.3390/ijms23169205_

Round 1

Reviewer 1 Report

In my opinion, the manuscript titled: “Epipharyngeal Abrasive Therapy (EAT) suppresses the secretion of IL-6 that is involved in the pathogenesis of various systemic diseases including IgA nephrophaty and Long COVID” presents an interesting research about the ability of continuous EAT to suppress the secretion of IL-6 and TNFα by preventing the collection of inflammatory cells in the epipharynx. I find particularly interesting that the EAT may contribute to the improvement of systemic inflammatory diseases associated with high expression levels of IL-6 : this finding  gives input to further research about  the potential use of EAT in the treatment of some diseases correlated with chronic epipharyngitis, like long COVID.

In general, the research is well conducted and well presented and the manuscript is well written. The introduction clearly presents the topic and the importance of the study. Both materials and methods are accurately described and the discussion is detailed.

Author Response

Dear Reviewer 1

Thank you very much for providing important comments. We are thankful for the time and energy you expended.

Response to Reviewer 1 Comments

Comments: In my opinion, the manuscript titled: “Epipharyngeal Abrasive Therapy (EAT) suppresses the secretion of IL-6 that is involved in the pathogenesis of various systemic diseases including IgA nephrophaty and Long COVID” presents an interesting research about the ability of continuous EAT to suppress the secretion of IL-6 and TNFα by preventing the collection of inflammatory cells in the epipharynx. I find particularly interesting that the EAT may contribute to the improvement of systemic inflammatory diseases associated with high expression levels of IL-6 : this finding  gives input to further research about  the potential use of EAT in the treatment of some diseases correlated with chronic epipharyngitis, like long COVID. In general, the research is well conducted and well presented and the manuscript is well written. The introduction clearly presents the topic and the importance of the study. Both materials and methods are accurately described and the discussion is detailed.

Response:   There have been several reports that epipharyngeal abrasive therapy (EAT) was effective in improving systemic inflammatory diseases, but the details of the mechanism of EAT remain unclear. In this study, we clarified part of the anti-inflammatory mechanism of EAT by focusing on IL-6 in the local epipharynx. We are planning further systemic analysis to see how the improvement of local inflammation by EAT affects the whole body.

Reviewer 2 Report

Introduction: Examples of other diseases that are associated with epypharingitis are needed.

“In fact, the high expression level of IL-6 has been reported to be involved in the development of IgA nephropathy, ME/CFS, and Long COVID [14–16]”- where is this expression?

MATERIAL AND Methods: Section 2.4 – how the tissue samples have been collected? How was expression analyzed – primers for in situ hybridization ets. More detailed description of ISH and IHC methods is needed.

Results: Figure 3 – It is not clear what do the pictures represent, what does the staining show – IL-6 or different cell types.

Figure 4 and Table 1 represent one and the same results. Figure 4b – how it was decided presence or absence of expression – probably this should be explained in Material and Methods section. It is not clear that the numbers in fig. 4b represent number of patients.

TNF-alpha – results as in figure 3 and 4b are missing for it.

Discussion: “This study focused on IL-6, which plays a central role in the human cytokine network,

and histologically investigated the effect of EAT on the IL-6 secretion in the epipharynx.”- ISH does not represent secreted molecules. IHC probably detects secreted molecules but authors should explain how do they understand IL-6 is visualized extracellularly.

Other similar studies should be cited.

“TNFα is known to be involved in the pathogenesis of rheumatoid arthritis and psoriasis [34,35], suggesting that the suppression of TNFα expression in the epipharynx associated with EAT may contribute to the improvement of systemic inflammatory diseases along with the suppression of IL-6 expression“ – this statement seems speculative. TNF-alpha is involved in variety of pathologies (the same way as IL-6). More explanation is needed to clarify this relationship. Other examples are also needed.

Conclusions: should be adjusted according to the results from the study and to be more punctual.

The title is speculative and misleading, as IL-6 is involved in pathogenesis of variety of conditions and diseases. That is why I would recommend revision of the title according to the results of the study.

Author Response

Dear Reviewer 2

Thank you very much for reviewing our manuscript and offering valuable advice.
We have addressed your comments with point-by-point responses and revised the manuscript accordingly.

Response to Reviewer 2 Comments

Point 1: Introduction: Examples of other diseases that are associated with epypharingitis are needed.

Response 1: Chronic epipharyngitis, like chronic tonsillitis, is considered a focal infection. Therefore, it has been reported that chronic nasopharyngitis induces various autoimmune diseases. We additionally described palmoplantar pustulosis, rheumatic arthritis, and psoriasis as other diseases associated with chronic epipharyngitis in Introduction section.

“Epipharyngitis is involved not only in otolaryngology disorders but also in the development of systemic diseases such as IgA nephropathy, palmoplantar pustulosis, rheumatic arthritis, psoriasis, myalgic encephalomyelitis/chronic fatigue syndrome (ME/CFS), and Long COVID [1, 5-9].”

Point 2: “In fact, the high expression level of IL-6 has been reported to be involved in the development of IgA nephropathy, ME/CFS, and Long COVID [14–16]”- where is this expression?

Response 2: We thank the reviewer for the careful review of the manuscript. It has been reported that circulating IL-6 is elevated in these pathologies. We have added this content to the text in Introduction section.

“In fact, the high expression level of IL-6 in serum or plasma has been reported to be involved in the development of IgA nephropathy, ME/CFS, and Long COVID [14-18].”

Point 3: MATERIAL AND Methods: Section 2.4 – how the tissue samples have been collected? How was expression analyzed – primers for in situ hybridization ets. More detailed description of ISH and IHC methods is needed.

Response 3: We added detailed descriptions in Methods section 4.1. and 4.4. Primer sequences of RNAscope are not publicly announced. Thus, we added the product numbers of each probe.

“Samples were obtained by endoscopic epipharyngeal biopsy between July 2021 and August 2021.”

“To detect IL-6 and TNF-α mRNA on the tissue, an RNA scope (in situ hybridization system, Advanced Cell Diagnostics, Hayward, CA, USA; IL-6: No.310371, TNF-α: No.310421) was used following the manufacturer’s guidelines.”

“All images were captured using microscope (AXIO Vert.A1, Carl Zeiss Inc.).”

Point 4: Results: Figure 3 – It is not clear what do the pictures represent, what does the staining show – IL-6 or different cell types.

Response 4: We thank the reviewer for the careful review of the manuscript. Positive signals of IL-6 RNA are showed as red dots. To recognize easier, we added arrow heads in Figure 2. Brown IHC signals are showing Protein expression of each cell marker. We also added name of targets (CD20: B cell, CD3: T cell, etc.) in Figure 2. 

Point 5: Figure 4 and Table 1 represent one and the same results. Figure 4b – how it was decided presence or absence of expression – probably this should be explained in Material and Methods section. It is not clear that the numbers in fig. 4b represent number of patients.

Response 5:  We changed Figure 3b to showing in percentage. Moreover, we added description of methods (section 4.4) how we determined the positivity.

“The positivity of signals in the subepithelial region was determined by pathologists at high magnification. Briefly, we randomly chose three areas at the subepithelial region and determined based on at least 300 cells.”

Point 6: TNF-alpha – results as in figure 3 and 4b are missing for it.

Response 6: We have not included a detailed analysis of TNF because we want the paper to focus on IL-6. We are considering reporting a separate paper with a more detailed analysis. If you think that additional analysis is necessary to TNF, we would appreciate an extension of the period of revision.

Point 7: Discussion: “This study focused on IL-6, which plays a central role in the human cytokine network,and histologically investigated the effect of EAT on the IL-6 secretion in the epipharynx.”- ISH does not represent secreted molecules. IHC probably detects secreted molecules but authors should explain how do they understand IL-6 is visualized extracellularly. Other similar studies should be cited.

Response 7: Immunohistochemistry of cytokines often poses a problem of non-specific binding. In this respect, ISH is highly specific and enables identification of expressing cells, so ISH was performed in this study. It has been reported that IL-6 mRNA and protein expression in lymphoid tissues are correlated. I added the relevant text to the text along with the references cited in Discussion section.

“In this study, we attempted to identify IL-6-secreting cells in the epipharynx by cytokine RNA in situ hybridization with a high degree of specificity [29].”

“Since mRNA expression of IL-6 in lymphoid tissues correlates with local protein expression [34, 35], the reduced mRNA expression associated with EAT is assumed to suppress IL-6 secretion in the epipharynx.”

Point 8: “TNFα is known to be involved in the pathogenesis of rheumatoid arthritis and psoriasis [34,35], suggesting that the suppression of TNFα expression in the epipharynx associated with EAT may contribute to the improvement of systemic inflammatory diseases along with the suppression of IL-6 expression“ – this statement seems speculative. TNF-alpha is involved in variety of pathologies (the same way as IL-6). More explanation is needed to clarify this relationship. Other examples are also needed.

Response 8: We agree with your assessment. TNFα is involved in various systemic diseases. Then, this paper focuses on autoimmune diseases associated with both TNFα and chronic epipharyngitis. There is a report that EAT was effective against some of these autoimmune diseases described, which is added to the text along with the cited references in the fourth paragraph in Discussion session. The reduction of TNFα mRNA expression in the epipharyngeal mucosa by EAT is considered to be one of the important mechanisms of EAT, and we are planning research targeting TNFα in the future.

Point 9: Conclusions: should be adjusted according to the results from the study and to be more punctual.

Response 9: We agree with your assessment. This study clarified that EAT subdued local inflammation in the epipharynx but did not confirm the effects on blood circulation or other organs. In order not to exaggerate the results of the study, we have adopted the results of the study that EAT induced IL-6 mRNA downregulation in the epipharynx as the Conclusion. In addition, what is expected from this result is described as a suggestion.

“EAT reduces the mRNA expression of IL-6, a key cytokine in chronic inflammation, in the epipharynx, suggesting that EAT may contribute to the improvement of systemic inflammatory diseases involving IL-6.”

Point 10: The title is speculative and misleading, as IL-6 is involved in pathogenesis of variety of conditions and diseases. That is why I would recommend revision of the title according to the results of the study.

Response 10: Similar to Conclusion, we simply adopted the results of the study that EAT induced IL-6 mRNA downregulation in the epipharynx as the Title.

“Epipharyngeal Abrasive Therapy (EAT) Reduces the mRNA Expression of major proinflammatory cytokine IL-6 in the epipharynx”